# Deep-Net: A Lightweight CNN-Based Speech Emotion Recognition System Using Deep Frequency Features

**DOI:** 10.3390/s20185212

**Published:** 2020-09-12

**Authors:** Tursunov Anvarjon, Soonil Kwon

**Affiliations:** Interaction Technology Laboratory, Department of Software, Sejong University, Seoul 05006, Korea; tursunovanvarjon@gmail.com (T.A.); mustaqeemicp@gmail.com (M.)

**Keywords:** artificial intelligence, deep learning, deep frequency features extraction, speech emotion recognition, speech spectrograms

## Abstract

Artificial intelligence (AI) and machine learning (ML) are employed to make systems smarter. Today, the speech emotion recognition (SER) system evaluates the emotional state of the speaker by investigating his/her speech signal. Emotion recognition is a challenging task for a machine. In addition, making it smarter so that the emotions are efficiently recognized by AI is equally challenging. The speech signal is quite hard to examine using signal processing methods because it consists of different frequencies and features that vary according to emotions, such as anger, fear, sadness, happiness, boredom, disgust, and surprise. Even though different algorithms are being developed for the SER, the success rates are very low according to the languages, the emotions, and the databases. In this paper, we propose a new lightweight effective SER model that has a low computational complexity and a high recognition accuracy. The suggested method uses the convolutional neural network (CNN) approach to learn the deep frequency features by using a plain rectangular filter with a modified pooling strategy that have more discriminative power for the SER. The proposed CNN model was trained on the extracted frequency features from the speech data and was then tested to predict the emotions. The proposed SER model was evaluated over two benchmarks, which included the interactive emotional dyadic motion capture (IEMOCAP) and the berlin emotional speech database (EMO-DB) speech datasets, and it obtained 77.01% and 92.02% recognition results. The experimental results demonstrated that the proposed CNN-based SER system can achieve a better recognition performance than the state-of-the-art SER systems.

## 1. Introduction

The affective content analysis of speech signals is an active area of investigation in this era. Speech is the greatest prevailing way to exchange information among human beings, and it is worth paying attention to human-computer interaction (HCI). On the other hand, social media, written text, and other semantic information methods are also very helpful and play a crucial role in HCI. The most significant factor in human speech is emotions, which can analyze for judgments about human expressions, paralanguages, and others. Hence, the speech signal is an efficient way for the fastest communication among HCI, which efficiently recognized human behavior. Today, emotion recognition in a speech signal is one of the fastest emerging research field, where researchers have developed methods to naturally detect emotions from a speech signal [1,2]. The theory of speech emotion recognition (SER) is beneficial for education and health, and it will be widely used in these fields once they are proposed [3]. The SER plays an important role in the HCI, and researchers are making a variety of techniques in the current decade to make it efficient and robust for real-time applications [4]. In the past decade, it has been a challenging task to recognize the emotional facts and the expressive cues from the speech of an individual due to the lack of techniques and technologies [5,6]. In each era, researchers have worked to develop an efficient SER system, and they have developed several methods for preprocessing, features extraction, and classification [7]. Some researchers have concatenated the low-level descriptor (LLD) with high-level statistical functions (HSFs) for emotion recognition tasks, and they have obtained high accuracy as compared to the other features in INTERSPEECH2013 [8]. In addition, many other researchers have developed a lot of new techniques for feature extraction. Turgut et al. [9] presented an idea for feature selection based on the definition set and the emotional state. Some researchers are considering the domain invariant and the emotional features for an efficient SER system that learn high-level salient and discriminative sensitive features from oral data [10]. Hence, the selection of optimal features for the SER is still challenging problems [7], and researchers are still continuously making efforts to achieve effective features for frames and utterances. The typical framework of speech emotion recognition system is illustrated in Figure 1.

With the development of skills and technologies, researchers have adopted artificial intelligence (AI) and deep learning (DL) approaches to enhance the way of the HCI, which includes emotion recognition in speech signals. Today, researchers have developed new techniques for efficient SER using AI and DL in this domain [6]. Many researchers have achieved great success in this era in order to make an efficient and robust SER system using certain DL applications, such as a deep belief network [11] (DBN), CNNs [12], and long short-term memory (LSTM) [13,14]. Most DL-based CNNs models work on 2D data, so researchers utilize deep spectrum approaches for the SER in order to extract high-level discriminative features from the speech spectrograms using the CNN model. All the CNNs models are extremely good working in the spatial domain to extract the salient features from the data [15]. Recently, fully convolutional networks (FCNs) have been introduced as a variant of the CNNs, which handle inputs of variable sizes based on their properties, and they have achieved state-of-the-art accuracy in time-series problems [16]. However, the FCNs model is not able to learn temporal features in this regard. The recurrent neural network (RNN) and the LSTM show good performances to model temporal dependency among the sequences [14,17]. The RNN-LSTM network is suitable to learn long term contextual dependencies, and it is widely used in the SER domain [18]. However, different CNNs models are proposed in the SER domain to improve the performance of the baseline SER system. The approach proposed herein aims to reduce the computational cost and improve the performance by using the lightweight CNN model.

In this study, we addressed two issues. The first issue is to reduce the cost complexity, and the second is to improve the state-of-the-art performance. Hence, we proposed a novel lightweight CNN model for the SER using plain rectangular kernels with a modified pooling strategy. Our model mainly focuses on the frequency features in the speech spectrograms in order to recognize the hidden emotional features by analyzing the deep frequency features. We used fewer parameters in our model to reduce the cost complexity. We tested our proposed system over two benchmarks, which included the interactive emotional dyadic motion capture (IEMOCAP) and the berlin emotional speech database (EMO-DB) speech datasets, to illustrate the significance and the robustness of the system. During the experimentation, our system secured high recognition results, so they were compared with the baseline model to show the effectiveness. We illustrate the qualitative and the quantitative results of the proposed system in Section 4. The key contributions of the proposed SER system are illustrated next.
We proposed a simple and lightweight convolutional neural network (CNN) architecture with multiple layers using modified kernels and pooling strategy to detect the sensitive cues based on the extraction of the deep frequency features from the speech spectrograms, which tend to be more discriminative and reliable in speech emotion recognition.We proposed new plain rectangular shape filters for the convolutions and the pooling strategy in order to extract the deep emotional features from the speech spectrogram utilizing frequencies’ pixels with reliable receptive fields to ensure and enhance the performance of the baseline SER models.We reduced the cost computation and the time complexity of the proposed SER model by utilizing fewer convolutional layers with small respective fields in order to learn discriminative features from the speech spectrograms, which was proven in the experimentations. For the convenience of the readers, we documented the statics in experiments and results section.We used a novel technique in this study for the SER and endorsed it as the recent success of the deep learning methods based on the deep frequency features collection that can increase the recognition rate of the SER compared with the state-of-the-art approaches’ results. The proposed technique still has not been used in the SER literature for feature collections. Thus, the usage of the frequency features collection enhances the model performance.We tested our suggested SER model on benchmarks, which included the IEMOCAP and the EMO-DB datasets. They achieved 77.01% and 92.01% recognition results. In the comparative analysis, our system showed outperformed recognition results that were proven from our experiments. Thus, our proposed SER system is the most suitable and applicable for monitoring the speakers’ emotions in real-time due to its simplicity.

The rest of the article is split into the following sections. Section 2 represents the related work of the SER, and Section 3 illustrates the proposed framework and its main components in detail. The experimental setup and the practical results of the suggested SER system are illustrated in Section 4, and Section 5 presents the discussion and a comparative analysis. The manuscript’s conclusion and a future direction for further research is presented in Section 4.

## 2. Materials and Methods

In the literature, the SER has been an active research area during this decade, and many techniques are proposed in this field for an efficient SER system that utilizes new technologies. Today, deep learning approaches are rapidly used in the SER due to the increase in data and high computational power. Most researchers are showing interest in deep learning-based methods over the traditional methods, because the deep learning method is the most dominant approach [2,15]. Hence, many researchers have used the CNN model to extract the high-level features in order to show the importance and the power of deep learning, which learn high-level discriminative features [19]. The CNN model achieved a lot of success in the visual recognition task, so researchers have therefore widely used CNNs for the feature representations in many speech analysis problems. For example, in Reference [20], the authors developed an SER system based on the CNN features using speech spectrograms. In addition, Reference [21] similarly used CNN to learn the salient features and recognized the emotions in spectrograms. Today, deep learning approaches have become a recent trend to extract the CNN features from the speech spectrum and the spectrograms [22]. Moreover, the researchers take benefits from the deep spectrum and the spectrograms by employing transfer learning techniques to trained end-to-end SER models utilizing Alex Net [23] and VGG [24]. The deep spectrum and the spectrograms are suitable for 2D representations of speech signals and show better performances to produce salient high-level features in the SER task, which can easily achieve the state-of-the-art results [25]. Furthermore, for temporal information, the researchers have used sequence learning methods, such as RNNs and LSTM networks to recognize the contextual dependencies [26]. In this regard, these networks are frequently adopted in the SER fields [27].

Through the improvement of technologies, artificial intelligence and CNNs are the most popular sources that have achieved excessive success in many fields, such as handwriting recognition [28], object recognition [23], natural language processing [29,30], and SER [31]. The convolutional neural networks addressed the scalability issues of the traditional neural networks [32,33] by allowing them to share similar weights for multiple regions of the inputs [34]. Usually, the CNN model consists of three main building blocks that first include the convolution layers, second the pooling layers, and finally the fully connected layers. The feature map of the input is computed by the convolution operation, and down sampling the dimension of the computed feature map is performed by the pooling strategy. Finally, the features are passed from the fully connected layers to be transformed into more reliable forms for the target predictions. Recently, CNNs has been used widely in the SER to learn the high-level discriminative features using the speech data [35] and classifying them accordingly [36]. The combination of the CNNs with the RNNs-LSTM is presented in Reference [37], and it improves the performance of the existence SER system.

Currently, researchers have used deep learning approaches for the SER. Ref. [21] proposed a new technique for the emotion recognition based on the DCNN. This method extracts features by using the Alex Net model and a trained conventional classifier, which is a support vector machine (SVM), to predict the emotions [38]. A CNN model extracts features from the whole utterance and feeds them to the LSTM or the RNNs to extract long term contextual dependencies in the speech signals [17]. Wen et al. [39] presented a method for the SER using the DBN and the SVM where the high-level features are extracted by the DBN and then classified by the SVM. Similarly, ref. [11] used the DBN model to represent the salient features and fed to the SVM for classification using the Chinese dataset built by Chinese Academy of Sciences’ Institute of Automation (CASIA) [40]. In Reference [2], the authors proposed a novel SER system based on a stride CNN to extract the salient discriminative features and trained them in an end-to-end manner. Furthermore, ref. [13] presented a SER based on the key segments to reduce the cost and increase the accuracy utilizing three challenging speech emotion datasets [41]. However, in this research, we proposed a novel CNN architecture for the SER based on the deep frequency pixel features to recognize the emotional features in the speech spectrograms. We used plain rectangular shape filters in the convolution layer with a modified pooling strategy to extract the hidden deep frequency features. In this research, we proposed a simple and lightweight CNN model for the SER using fewer convolutional layers in order to reduce the cost complexity and improve the performance of the baseline methods. A detailed explanation about the proposed SER is illustrated in the subsequent section.

## 3. Proposed Approach of the SER

In this section, we present a detailed overview of the suggested frameworks of the SER and its related components. We utilized a simple and lightweight end-to-end CNN model in this study to learn the high-level salient features from the speech spectrograms. The speech spectrogram is one of the suitable 2D representations of a speech signal that is mostly used as an input of the CNN models for a speech analysis. The proposed CNN architecture which consists of two main phases, the spectrograms, plotting the frequencies of the timing varying speech signals, and the CNN model, to learn deep frequency features and classify them accordingly. A detailed explanation of the spectrogram generation and the CNN model is presented in the subsequent sections.

### 3.1. Generation of the Speech Spectrograms

A spectrogram is a two-dimensional representation of the speech signals, and a dimensional setting of the speech data for 2D deep learning models is a challenging task. The dimension conversion of the speech data was necessary for the 2D CNN model to learn the salient discriminative features from the speech signals to make an efficient SER system. A spectrogram is 2D representations of the speech data, which is suitable to show the strength of the speech signals with respect to the frequencies [4].

For a visual representation, we made a spectrogram to plot the speech signals with respect to time by applying a short-term Fourier transformation (STFT) algorithm. The STFT algorithm is used to convert the short-term speech signals or the long-time speech signals into segments, which have similar lengths. Finally, the fast Fourier transformation (FFT) is applied to that segment to compute the Fourier spectrum. Actually, the spectrogram is a combination of the frequencies with respect to time in two dimensions. The x-axis of the spectrogram shows the time *t,* and the y-axis illustrate the frequency range *f* of each speech signal, which is computed as:(1)$$X\left(\tau ,\omega \right)=\int_{-\infty }^{\infty }S\left(t\right)f\left(t-\tau \right)e^{-i\omega t}\;dt.$$

For example, the speech spectrogram *S* has different types of frequencies over various time contains *S* (*t*, *f*) of the corresponding signal. In the spectrogram, the frequencies are represented by the colors. The dark color shows a low magnitude frequency, and the light color shows a high magnitude frequency. In this decade, many researchers recommend a spectrogram for a 2D representation, which is the most suitable for the CNN model to learn the high-level features for an efficient SER system [42]. Some speech spectrogram samples are illustrated below in Figure 2.

### 3.2. Proposed SER Model Architecture

The suggested CNN architecture is illustrated in Figure 3. Most CNN models are designed for different types of tasks that are related to computer visions, such as classification, localization, recognition, tracking, and retrieval in order to achieved high accuracies [43,44]. We were inspired by their performance and adapted it into the SER domain to make a significant and robust system to recognize emotions in the speech data. Basically, the CNN model has three components, which include the convolution layers, the pooling layers, and the fully connected layers. The convolution layers have many numbers of filters to compute the initial feature map and then feed to the pooling layer in order to reduce the dimensions for fast processing and computations. Similarly, the fully connected layers are used to learn the global features and recognize the hidden cues in the input data and then fed to the SoftMax layer to produce the different probabilities of classes. In this analysis, we utilized the basic structure of the CNN and constructed a simple and lightweight model for the SER using plain rectangular shape filters in the convolution layer with a modified pooling strategy. Our model used the same structure filters in the convolutional layer and in the pooling layer with a small receptive field in order to capture the hidden cues and the important cues. Our model uses the simple rules of CNN architecture, which means that the number of filters must be the same when the same output features map is required. If the size gets cut in half or is reduced up to a half, then the number of filters must be double in the convolution layer to sustain the complexity in each layer. The visual representation of the proposed model is illustrated below. Detailed model layers, input, output and parameters are given in Table 1.

We used a simple and lightweight CNN model to extract the high-level salient features utilizing deep frequency pixel after obtaining the speech spectrograms. Actually, the CNN signifies the feedforward neural networks, which have multiple layers, such as convolution, pooling, and fully connected layers, that exploit the local connectivity and the correlation among the neurons in the connecting layers. Our designed CNN model consists of eight convolution layers with a rectified linear unit (ReLU), three max-pooling layers followed by batch normalization (BN), and two fully connected layers with a SoftMax classifier [24]. The following are detailed explanations of each layer.

The first convolution layer (C1) has ten filters that are a (9 × 1) size to prepare the data or the features map by utilizing ten (10) filters that produce a new tensor deprived of padding. In the second layer, C2 has ten filters, which are size (5 × 1), that evaluate the obtained features map of C1 resulting from the new tensor. The C2 layer focuses on the similarities and measure them accordingly. We used a modified pooling strategy in this model that followed the convolutional layers. We used a more convolutional layer to make deeper networks to extract the high-level salient features. Similarly, the C3 layer has also the same numbers of filters, which are a (3 × 1) size, using no stride setting and padding that obtain a new features map that has the same size as C2. After C3, we used a max-pooling layer of filter size (3 × 1) with no stride setting to reduce the dimensionality [24]. The pooling layers always added on the top of the convolution layer to compute the resolutions and represent the subsampling. The pooling layer followed by the BN layer to rescale or normalize the data to increase the performance of the model and reduced the cost computations.

After the normalization of the feature maps and the pooling layer, the number of filters must be increased in the convolution layer to obtain the discriminative features. Furthermore, in C4, we increased the number of filters up to 40, and the size of each filter was (3 × 1) with no stride setting and padding. C4 uses the normalized features and produces a new tensor for the next layer. Similarly, C5 utilizes the same number of filters with the same size in the convolutional operations without stride and padding to extract the most hidden cues using the frequency pixels. After the C5 layer, we again used the pooling strategy with a new filter size to reduce the dimensionality and represent the subsampling. Here we used a (2 × 1) filter in the pooling with no stride setting, which is again followed by the BN layer to rescale and normalize the input features map, which increases the performance of the model. Similarly, the C6 layer yields normalized to input and extract the deep features by utilizing 80 filters that are a (10 × 1) size using no stride and no padding. The C6 layer is followed by the C7 layer, which has the same number of filters, but the filter size is different. The C7 layer utilizes a (1 × 1) filter size with zero strides and zero padding to extract the high-level cues with the same dimension to improve the performance. After the C7 layer, we utilized another pooling layer with a (2 × 1) size, which was followed by the BN layer to perform the subsampling and rescaling.

In the last convolution layer C8, we used 80 filters with size (1 × 1) using zero strides and padding to improve the features and the performance of the model. In the designed CNN model, we used three pooling layers that included 3 × 1, 2 × 1, and 2 × 1 filter sizes, to reduce the spectral dimensionality and preserve more information that is useful for the emotion recognition. The output tensor of C8 is fed into the fully connected layers to recognize the global information. We used two fully connected (FC) layers with 80 neurons and 30 neurons. The FC layer represents the traditional neural network and combines it with a convolutional layer in the CNN to make it end-to-end architecture using a SoftMax classifier and label information to produce the different probabilities of the classes.

However, a CNN model contains a two-step training process, like the traditional neural networks, such as forward propagation and backward propagation. The main aims of these two steps are to compute the results of the input data with the current parameters and update the trainable parameters. The update parameters again computed the results and showed their performance. Our designed CNN model utilizes frequency pixels to extract the high-level features, and it recognized emotions in the speech data using the speech spectrograms. Our model used a novel filter shape in the pooling and the convolution operations, which capture the deep features from the frequencies pixels and measure the correlations with an adjacent layer. Our model generates a robust result with a simple and lightweight model by utilizing this novel architecture, which has fewer layers to reduce the cost complexity, and it was proved in the experimentations.

### 3.3. Proposed Model Configuration

We implemented the proposed CNN architecture in python utilizing a machine learning library called scikit-learn and other related supporting libraries. The model was trained on the speech spectrograms, which are generated from the speech signals. We converted each audio file into an individual spectrogram, which had a size of 64 × 64 pixels, to give it as an input to the model to extract the high-level deep frequency features. We divided the generated data into two folds that included training and testing with a 70: 30 ratio. We utilized 70% of the data for training purposes, and the remaining 30% of the data was used for model validating. We used a 5-fold cross-validation method to test the model generalization. In this method, we used the machine learning library in python to divide our dataset into five-folds and train the models accordingly.

We used a single NVIDIA GeForce GTX 1070 GPU with eight gigabytes onboard memory for the model training and the validating process. We trained the model for 100 epochs with a 0.00001 learning rate and selected decay one unit after 10 epochs. We tried various batch sizes during the model training, but the best result was achieved on 256, which was finally selected for the whole model training process. The Adam optimizer we used in the model training secured the best accuracy for the SER with 0.333 training lost and 0.55 validation lost. Our model architecture is very simple, and it takes less time with training due to its compact model size, which indicates the model’s simplicity and shows the ability for real-time SER applications.

## 4. Experimentations and Results

We practically prove our system in this section, which we tested over two benchmark IEMOCAP [45], and EMO-DB [46] speech datasets in order to show the significance and the robustness of the model for the SER using speech spectrograms. We checked the performance of our SER system and compared it with other baseline SER systems using the same phenomena. A detailed description of the utilized datasets, the accuracy matrices, and the experimental results are discussed in the upcoming sections.

### 4.1. Datasets

The IEMOCAP, which is the interactive emotional dyadic capture, is the speech emotion acted dataset consuming the English dialogs. The dataset has five sessions, and each session has two speakers, which include one male and one female to record the different emotions. There is a total of 10 speakers that were used to record the dataset, and all the speakers are professionals or experts. The dataset contains 12 h of audiovisual data that was recorded with a 16 kHz sampling rate. There are different emotions, and all the emotions were scripted, which were recorded by professional actors using two types of scripts that included a scripted version and an improvised version. The details of the emotions, the total samples, and the participation rate in percentage are illustrated in the Table 2.

In the EMO-DB, which is the Berlin emotional speech database, five female and five male actors recorded ten sentences for each of the seven emotions. 535 pieces of German speech audio data were recorded in different emotional states. These audio files in wave formats contain speech signals sampled at 16 kHz rate, 16-bit size, 256 kbps bit rate, and they are approximately 2–3 s long. The details of emotions, total samples, and participation rate in percentage are illustrated in Table 3.

### 4.2. Experimental Evaluation and Results

The test set is used to evaluate the model recognition capability on the unknown utterances of data. The error in the model prediction approximates the system generalization error [47]. In this paper, the cross-validation estimation method is utilized to evaluate all the datasets in an effective manner. The data in the database is split into two parts, which include the training data and the testing data. The initial data is separated into k parts. One part is used as a test data, and the other k data is used for training. Each part is then used diagonally for testing. The test operation, which is repeated k-times, in different parts of all the data is called a k-fold cross-validation [48]. We used a well-known 5-fold cross-validation assessment method for the analysis of our method.

We utilized the statistical parameters to investigate the accomplishment of the proposed method, which are computed using the formulas [36]. The True-Positive (TP) which is the number of correctly predictable real positive records, False-Negative (FN), which is the number of wrongly predictable real positive records, True-Negative (TN), which is the number of correctly predictable real negative facts, and the False-Positive (FP), which is the number of wrongly predictable real negative numbers [49]. T describes correct predictions, F describes wrong predictions, and the summation of the TP and the FN is shown as the amount of positive data. The summation of the TN and the FP indicates the amount of negative data in the real condition [50]. The accuracy identifies the overall recognition ratio. Meanwhile, the accuracy alone will not be satisfactory, and additional parameters, such as precision, recall, and the F1-score evaluation criteria are required. The F1-measure is defined as the harmonic mean of the precision rate and the recall rate [51]. Additionally, we evaluated our proposed system with different perspectives and show their results in terms of the weighted accuracy and the un-weighted accuracy, which is used in the literature, and we used them for our comparative analysis. The training performance of our system is illustrated in Table 4.

Table 4 shows the model training results for the IEMOCAP and the EMO-DB datasets that utilize the speech spectrograms as the input features. We obtained high accuracy for both databases. In this study, we used a novel CNN model for the SER using modified filters in convolutions and pooling strategies. Our model increases the level of accuracy and also decreases both losses and the training and testing, which indicates the significance and the effectiveness of the model. The visual result of the proposed CNN model is illustrated in the Figure 4, which shows the training and the validation accuracies, as well as the loss, for both datasets.

### 4.3. Model Predictions Performance

We used acted databases, such as IEMOCAP and EMO-DB that have different emotions, which were chosen according to Ekman’s theory. We evaluated the proposed model and show their prediction performance in the Table 5 and Table 6 and the confusion matrices obtained. Our model shows the precision, the recall, the F1-Score, the weighted results, and the un-weighted results in the prediction performance, which clearly indicates the model robustness over the state-of-the-art methods.

The Table 5 and Table 6 illustrate the classification reports of the IEMOCAP and the EMO-DB databases to the precision and the recall values for every emotion in the dataset. Our model shows better recognition results than the state-of-the-art methods. The overall performance of the proposed system is the significance, and it outperforms for all the emotions even happiness [13]. The baseline models recognized the happy emotion with low accuracy due to their linguistic information, but our model recognizes the emotions from the frequency pixels in order to address and overcome the recognition accuracy and the overall model cost computation. That is why our SER model recognizes all the emotions with a better recognition rate. Hence, for further investigations to show the class-wise accuracy and the confusion among each other, we need to obtain the confusion matrix for each dataset which is shown in Figure 5. The confusion matrix shows the actual predicted values diagonally and the confusion among each other in corresponding rows. The confusion matrixes are illustrated below for assistance.

According to the IEMOCAP confusion matrix, the highest results obtained for anger, sadness, and neutral were 85%, 74%, and 81% respectively. The happy emotions were also bitterly recognized as compare to state-of-the-art methods, and the overall recognition results of the proposed model are better than baseline methods. In the EMO-DB experiments, the highest accuracy results of 96%, 99%, 99%, 93%, and 92% were obtained for anger, boredom, disgust, fear, and sadness, respectively. The lowest accuracy result, which was 78.87%, was achieved for happiness. For the F1-measure, the highest results were 95.16%, 92.72%, and 88% for neutral, and the other lowest result, which was 80%, was for happiness. The high precision rates of 96%, 99%, 99%, 93%, and 92% were for anger, boredom, disgust, fear, and sadness, respectively, and the lower precision rates of 88% and 80% were obtained for neutral and happiness. Our model recognized all these emotions with a better recognition rate and a reduced model size or cost computation due to their structure and architecture. Our model utilized simple architecture and user-friendly working features, which is capable of real-time applications to monitor human behavior. The training performance of our proposed SER system is illustrated in Table 7 in order to show the model complexity, the time, the size, and a comparison with other baseline SER systems.

Table 7 shows the model’s entire training time and recognition accuracy where the comparative analysis indicated that the proposed SER system is better and cost-friendly. In this table, we compare our model with the other SER models. The time complexity and the recognition accuracy of our model is less than the other baseline systems for both speech corpus. Our model has a simple structure to use fewer layers with a novel filter shape and a modified pooling strategy in order to recognize emotions from the speech spectrograms.

## 5. Discussion and Comparative Analysis

In this study, the proposed CNN architecture, which is a plain rectangular shape convolutional filter, and a modified pooling strategy are considered as a major contribution for the SER. We used a novel filter shape and recognized the deep frequencies features from the speech spectrogram by applying a modified pooling technique in order to reduce the dimensionality and increase the accuracy. According to the best of our knowledge, the SER domain lacks this type of filter and frequency feature selection to recognize the emotions in the speech data. We deeply investigated the literature of the SER domain, which has two major issues that include i: the recognition rate and ii: the cost computations or the time complexity. The researchers have developed many techniques in this domain utilizing hand-crafted features and high-level features using the classical models and the CNN models, but the recognition accuracy is still low, and the cost complexity is very high. In this research, we addressed these issues and made a novel CNN model for the SER using a modified filter shape and a pooling scheme in order to recognize the emotions in the speech data. Our model used a speech spectrogram as the input, which is a visual representation of the frequencies. We designed a new filter for the convolutional operation with a plain rectangular shape to extract the deep CNN features from the frequency pixels, which were recognized accordingly. With the usage of the deep frequency features, our model recognized the speech emotions with high accuracy. After resolving this issue, we focused on time complexity, which followed the designed filter and a pooling scheme, and made a simple CNN model that utilized fewer layers. Our proposed model used eight convolutions, three poolings, 2 batch normalizations, and 2 fully connected layers with a SoftMax classifier in order to recognized the emotions in the speech. Due to this simple structure, we reduced the time complexity and made an efficient and lightweight SER system, which was proved in the experimentation. See Table 6 for further assistance. We compared our results and model performance with the baseline models to show the significance and the robustness of the proposed system. A detailed overview about our comparative analysis is illustrated in the Table 8.

The above table represents the comparative analysis of the suggested SER system with other state-of-the-art baseline SER systems using the same speech corpus. In the table, we show the outperforming results of the proposed system, which is significantly higher than other systems, which indicates the effectiveness of our system. The proposed technique illustrates the recent success of deep learning in 2020 in the SER domain, which recognized all the emotions with high accuracy that includes even the happy emotion using a simple architecture. The recognition rate and the time complexity of the proposed SER model is better than the baseline model. A detailed demonstration and the experimental results are given in Table 7 and Table 8. In this study, we developed a simple and lightweight CNN model for the SER with high accuracy and reduced cost complexity, which is suitable for monitoring the real-time SER applications.

## 6. Conclusions and Future Direction

In this paper, a novel and lightweight CNN model was proposed for the SER based on the deep frequency features using speech spectrograms. A plain rectangular shape filter was used to extract the deep frequency features in order to ensure the accuracy. We designed a CNN model that utilized eight convolutional, three max-pooling, and two fully connected layers for the SER. Due to the usage of fewer layers, we reduced the cost and the time computation. The model performance was evaluated over two benchmarks, which include the IEMOCAP and the EMO-DB datasets, in order to show the effectiveness and the significance. The prediction results of the proposed system outperformed the state-of-the-art methods. Our model achieved outstanding 77.01% recognition results for the IEMOCAP dataset and 92.02% for the EMO-DB dataset. The proposed model increased by 5% and 6% over the state-of-the-art accuracy, which is a great achievement in the current era to recognize all the emotions with better accuracy and a reduced model size in order to show computational friendly output. The output of the proposed system showed the strength of the system for a SER that utilizes speech spectrograms.

In the future, we will focus on adopting this type of system in an automatic speaker recognition system and further elaborate it for signals processing tasks. Similarly, it would be advisable to test the strength of the system and the achieved outcomes on diverse databases, which include natural databases. The recognition rates can be increased and even combined with other deep learning methods. The studies can be enriched by both spectrums and spectrograms.

## Figures and Tables

**Figure 1 sensors-20-05212-f001:**
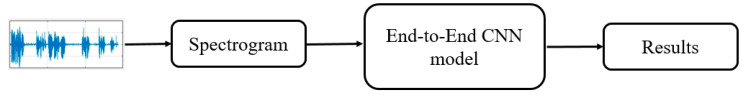
Typical flow of speech emotion recognition system, which is show the flow from raw data to classification results.

**Figure 2 sensors-20-05212-f002:**
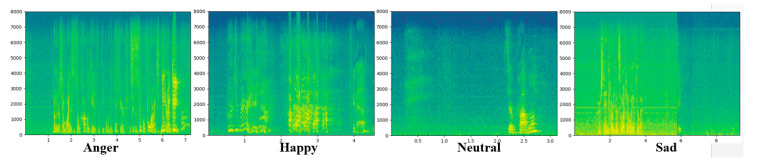
Spectrograms samples of different emotional speech signals.

**Figure 3 sensors-20-05212-f003:**
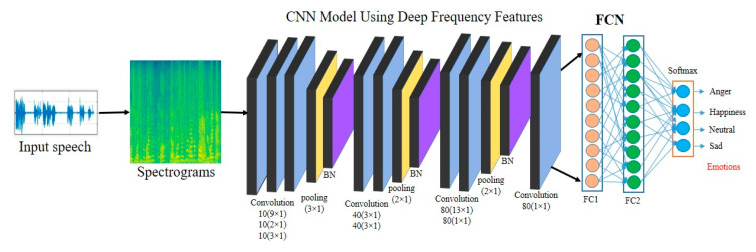
An overview of the proposed speech emotion recognition (SER) model: We proposed new plain rectangular shape fitters with a modified pooling strategy to learn and capture the deep frequency features from the speech spectrogram.

**Figure 4 sensors-20-05212-f004:**
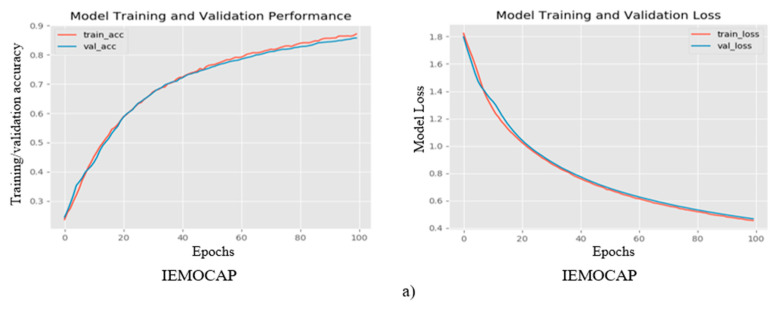
The figure shows the training performance of the proposed system and illustrates the training and the validation accuracies, as well as the losses, for the benchmark (**a**) IEMOCAP and (**b**) EMO-DB datasets.

**Figure 5 sensors-20-05212-f005:**
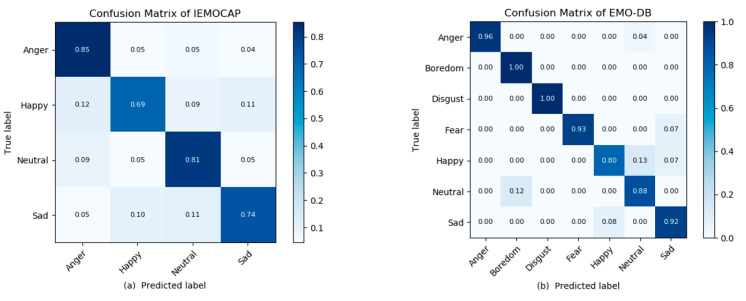
Confusion matrixes of the IEMOCAP and the EMO-DB datasets. (**a**) The IEMOCAP confusion matrix; (**b**) the EMO-DB confusion matrix. The x-axes show the predicted labels, and the y-axes show the actual labels.

**Table 1 sensors-20-05212-t001:** The overall specification of the proposed model is illustrated with set of convolutions layers, input and output size, and number of units which consist of kernel size, and number of parameters.

Model Layers	Input Tensor	Output Tensor	Parameters
Layer 1, 2, and Layer 3.	3 × 64 × 64	10 × 64 × 64	L1 = 280, L2 = 210, and L3 = 310
Pooling_1	10 × 64 × 64	10 × 62 × 64	0
Layer 4, and Layer 5.	10 × 62 × 64	20 × 62 × 64	L4 = 620, and L5 = 1220
Pooling_2	20 × 62 × 64	20 × 31 × 64	0
Layer 6, and Layer 7.	20 × 31 × 64	40 × 31 × 64	L6 = 2440, and L7 = 4840
Pooling_3	40 × 31 × 64	40 × 15 × 64	0
Layer 8.	40 × 15 × 64	80 × 15 × 64	L8 = 7260
Dense_1	35,840	80	2,867,280
Dense_2	80	30	2430
Softmax	Probabilities	Number of classes	Accordingly
Total parameters = 2,974,367			
Trainable parameters = 2,973,887			

**Table 2 sensors-20-05212-t002:** The detailed description about the interactive emotional dyadic motion capture IEMOCAP dataset about the emotions, the numbers of utterances, and the participation rates in percentage.

Emotions	Samples	Contribution in (%)
Anger	1017	19.94
Sadness	1120	19.60
Happiness	1636	29.58
Neutral	1650	30.88

**Table 3 sensors-20-05212-t003:** The detailed description about the berlin emotional speech database EMO-DB dataset in term of the emotions, the numbers of utterances, and participation rates in percentage.

Emotions	Samples	Contribution in (%)
Anger	127	23.74
Sadness	62	11.59
Happy	71	13.27
Neutral	79	14.77
Disgust	46	8.60
Fear	69	12.90
Boredom	81	15.14

**Table 4 sensors-20-05212-t004:** Training evaluation of the proposed model using the IEMOCAP and the EMO-DB speech datasets.

Input Feature	Dataset	Weighted Acc %	Un-Weighted Acc %	Accuracy %
Speech-Spectrogram	IEMOCAP EMO-DB	83 93	82 95	83 95

**Table 5 sensors-20-05212-t005:** Prediction performance of the suggested SER system for the IEMOCAP dataset using speech spectrograms.

Emotions/Classes	Recall Values	Precision Values	F1-Score
Anger	0.85	0.68	0.76
Sad	0.74	0.72	0.75
Happy	0.69	0.85	0.73
Neutral	0.81	0.78	0.80
Weighted Acc	0.77	0.76	0.77
Un-weighted Acc	0.76	0.76	0.76
**Accuracy**	**-**	**-**	**0.77**

**Table 6 sensors-20-05212-t006:** Prediction performance of the suggested SER system for the EMO-DB dataset using speech spectrograms.

Emotions/Classes	Recall Values	Precision Values	F1-Score
Anger	0.96	0.99	0.98
Sadness	0.92	0.86	0.89
Happiness	0.80	0.92	0.85
Neutral	0.88	0.82	0.86
Boredom	0.99	0.89	0.94
Fear	0.93	0.99	0.96
Disgust	0.99	0.98	0.97
Weighted Acc	0.93	0.93	0.93
Un-weighted Acc	0.92	0.93	0.92
**Accuracy**	**-**	**-**	**0.93**

**Table 7 sensors-20-05212-t007:** The comparison of the proposed model in term of time complexity and accuracy using the suggested speech datasets with the baseline SER methods.

Models	IEMOCAP	Accuracy	EMO-DB	Accuracy
ACRNN [52]	13,487 s	64.74%	6811 s	82.82%
ADRNN [53]	13,887 s	69.32%	7187 s	84.99%
CB-SER [13]	10,452 s	72.25%	5396 s	85.57%
**Proposed SER**	**3120 s**	**77.01%**	**1260 s**	**92.02%**

**Table 8 sensors-20-05212-t008:** Comparative analysis of the suggested SER system and the baseline SER system using the IEMOCAP and the EMO-DB speech corpus.

Authors/Reference/Year	Dataset	Un-Weighted Accuracy
Zhao et al. [37] (2019)	IEMOCAP	52.14%
Fayek et al. [54] (2017)	//	64.78%
Guo et al. [55] (2019)	//	57.10%
Zhang et al. [56] (2015)	//	40.02%
Han et al. [57] (2014)	//	51.24%
Meng et al. [53] (2019)	//	69.32%
Zhao et al. [58] (2019)	//	66.50%
Luo et al. [59] (2018)	//	63.98%
Jiang, S, et al. [60] (2019)	//	61.60%
Chen et al. [52] (2018)	//	64.74%
Issa et al. [61] (2020)	//	64.03%
Mustaqeem et al. [13] (2020)	//	72.25%
**Proposed Method (2020)**	**//**	**77.01%**
Guo et al. [55] (2019)	EMO-DB	84.49%
Meng et al. [53] (2019)	//	88.99%
Chen et al. [52] (2018)	//	82.82%
Badshah et al. [4] (2019)	//	80.79%
Jiang et al. [62] (2019)	//	84.53%
Issa et al. [61] (2020)	//	86.10%
Mustaqeem et al. [13] (2020)	//	85.57%
**Proposed method (2020)**	**//**	**92.02%**

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
