# Peer review of "Deep-Net: A Lightweight CNN-Based Speech Emotion Recognition System Using Deep Frequency Features"

_sensors, 2020, doi:10.3390/s20185212_

Round 1

Reviewer 1 Report

Dear authors ,

With interest we were reading the current version of the paper. We have the following critical remarks:

  1. The authors refere to [82], [83] and [84 in line 339 , but we were unable to find the references
  2. 2. In line 362 the authors refer to figures 14, 15 and 16, but we were unable to find them
  3. In the introduction line 32, 34 the authors stress that speech is the greatest source of exchanging informations. The authors don't consider written text and semantic. In social media e-mails etc written text has much more impact in social interactions.
  4. In several tables the authors compare the performance of SER systems. In table 6 it is not clear how the models are implemented.
  5. The authors a found a better performance of their systems using plain rectangular kernels  and a modified pooling strategy. Many other authors used similar ideas with worse performance. Why is the current model more successful.

Author Response

Response to Reviewer 1 comments

  1. The authors refer to [82], [83] and [84 in line 339, but we were unable to find the references.

Response: We are very grateful to point out the irrelevant reference numbers. We update and correct the references in the revised version. The updated text is given below for the ease of the reviewer.

”The TP simply represents True-Positive, which is the number of correctly predictable real positive records, the FN indicates False-Negative, which is the number of wrongly predictable real positive records, the TN shows the True-Negative, which is the number of correctly predictable real negative facts, and the FP illustrates False-Positive, which is the number of wrongly predictable real negative numbers [45]. T describes correct predictions, F describes wrong predictions, and the summation of the TP and the FN is shown as the amount of positive data. The summation of the TN and the FP indicates the amount of negative data in the real condition [46]. Accuracy identifies the overall recognition ratio. Meanwhile, the accuracy alone will not be satisfactory, and additional parameters, such as precision, recall, and the F1-score evaluation criteria are required. The F1-measure is defined as the harmonic mean of the precision rate and the recall rate [47].”

  1. In line 362 the authors refer to figures 14, 15, and 16, but we were unable to find them.

Response: Thank you to find out the points, which are not clear that’s needs attention. We update the figure numbers accordingly and the updated text is given below for the ease of reviewer and readers.

“We evaluated the proposed model and show their prediction performance in the tables below and the confusion matrices obtained, which are given in Figure 5.”

Justification: In the initial draft we placed the confusion matrices separately for both datasets and the final version, after our professor review we combine the confusion matric of both dataset and placed in single figure and forget the updating of figures numbers in a manuscript that’s why it’s mismatched from each other. Now in the revised version, we update it accordingly. Thanks for your extensive revision and sorry for our mistake.  

  1. In the introduction line 32, 34 the authors stress that speech is the greatest source of exchanging information. The authors don't consider the written text and semantic. In social media e-mails etc. written text has much more impact on social interaction.

Response: Thank you for your suggestion, you are right the text and semantic is also a better source of exchanging information and its play an important role in social interactions. We added some sentences about these sources but we showed the empowerment of speech signals in this study. The updated text of the revised version is given below for the ease of reviewer and readers.

The affective content analysis of speech signals is an active area of investigation in this era. Speech is the greatest prevailing cause to exchange information among human beings, and it is worth paying attention towards human-computer interaction (HCI). In the other hand, social media, written text, and other semantic information is also very helpful and play a crucial role in HCI. The most significant factor in human speech is emotions, which can analyze for judgments about human expressions, paralanguages, and others. Hence, the speech signal is an efficient way for the fastest communication among HCI, which efficiently recognized human behavior.”

  1. In several tables, the authors compare the performance of SER systems. In table 6 it is not clear how the models are implemented.

Response: We are very grateful to you to find out the points, which are not clear that’s needs more explanations. In this table, we just compare the training time of our system including accuracy on EMO-DB and IEMCAP speech datasets and compared it with some state of the art SER system. The actual comparison of the proposed system has been done in table 7, which has complete info about the system comparison. We added the explanation of table 6 in the updated version for the ease of reviewer and readers.

Table 6 shows the model’s entire training time and recognition accuracy where the comparative analysis indicated that the proposed SER system is better and cost-friendly. In this table, we compare our model with the other SER models. The time complexity of our model is less than the other baseline systems for both speech corpus. Our model has a simple structure to use fewer layers with a novel filter shape and a modified pooling strategy to recognize emotions from the speech spectrograms.”

  1. The authors found a better performance of their systems using plain rectangular kernels and a modified pooling strategy. Many other authors used similar ideas with worse performance. Why is the current model more successful?

Response: Thank you for your interest. The shape of our kernels is different than other researchers. In this study, we used a rectangular kernel which has cover only one column or cover-up only one frequency pixel at a time. Similarly, apply to pooling operations. Our main goal is to extract frequency info because the speech spectrogram is a visual representation of frequencies with respect to time. Furthermore, we used a novel CNN architecture with a new convolutional operation and pooling. That’s why our system presents better performance. You can see our basic structure, according to our knowledge, this is a new structure and neither used before in the speech emotion recognition domain. 

Reviewer 2 Report

The paper proposes a DL approach that has high accuracy and low complexity for SER. It contributes to the field of SER and/or DL.

Regarding the current manuscript, my comments are the following.

1. For "Deep-Net" in the title, how is it from, or what does it stand for? This name looks very general. Besides, "light weight"=>lightweight
2. Fig.1 citation is missing from the main text. In addition, the caption and the figure content of Figure 1 do not match.
3. L136-L138. Besides the tasks of objection recognition, NLP and SER introduced in the paper, convolution/filtering can also be used in image encryption (such as 10.1016/j.ins.2017.02.036 and 10.3390/e21030319). The authors may introduce or cite this work to show the power of convolutional operations.
4. It would be better to give the mathematical equations of STFT to describe its theory in Sec. 3.1.
5. What DL software platform/programming language does the paper use?
6. How does the proposed model contribute to reducing computational complexity? Why fewer convolutional layers with small respective fields can still learn discriminative features? How to decide the tradeoff between the accuracy and complexity of the proposed method?
7. L297, the numbers of the references IEMOCAP and EMO-DB are not displayed correctly.
8. The paper may calculate or list the numbers of the weights or parameters of the proposed CNN model to indicate its lightweight.

Some minor issues:
L95 ... to learn deep the discriminative features ...
L284 ... a 5-folds cross-validation method...

Author Response

Response to Reviewer 2 comments

  1. For "Deep-Net" in the title, how is it from, or what does it stand for? This name looks very general. Besides, "light weight"=>lightweight.

Response: We are very grateful to you for your suggestion to highlight the typo. The Deep-Net mean deep network or deep learning-based network. We utilized a deep learning strategy, that’s why we placed at the start. The other reason it looking good and make a sense according to the title. If you suggest removing so we can remove it no problem. Furthermore, sorry for the typo it’s lightweight not light weight. We correct it in the revised version.

  1. 1 citation is missing from the main text. In addition, the caption and the figure content of Figure 1 do not match.

Response: Thank you to find out the points, which are not clear that need attention. We properly cite the figure in the main text and update the figure caption accordingly and the updated text of the revised version is given below for the ease of reviewer and readers.

Hence, the selection of optimal features for the SER is still challenging problems [7], and researchers are still continuously making efforts to achieve effective features for frames and utterances. The typical framework of speech emotion recognition system is illustrated in Figure 1.”

Typical flow of speech emotion recognition system, which is show the flow from raw data to classification results.

  1. L136-L138. Besides the tasks of objection recognition, NLP and SER introduced in the paper, convolution/filtering can also be used in image encryption (such as 10.1016/j.ins.2017.02.036 and). The authors may introduce or cite this work to show the power of convolutional operations.

Response: Thank you to suggested references our work will be stronger by citing this work. The updated text from the revised version is below for easy to the reviewer.

The convolutional neural networks addressed the scalability issues of the traditional neural networks [32, 33] by allowing them to share similar weights for multiple regions of the inputs [34].”

  1. It would be better to give the mathematical equations of STFT to describe its theory in Sec. 3.1.

Response: We are very grateful to you to find out the points, which are not clear that’s need more explanations. We added the mathematical equation of STFT in the revised version. Now the revised manuscript is looking good and complete having no confusion. Thanks for your extensive review. Text from the updated version is below for the ease of reviewer and readers.

The x-axis of the spectrogram shows the time t, and the y-axis illustrate the frequency range f of each speech signal, which is computed as:

                                                        (1)

For example, the speech spectrogram S has different types of frequencies over various time contains S (t, f) of the corresponding signal.”

  1. What DL software platform/programming language does the paper use?

Response: Thank you for your interest. We deeply explain the whole process of our model configuration at the last of section 3, proposed methodology. The updated text from the revised version is mention below for easy to the reviewer.

We implemented the proposed CNN architecture in python utilizing a machine learning library called scikit-learn and other related supporting libraries. The model was trained on the speech spectrograms, which are generated from the speech signals. We converted each audio file into an individual spectrogram, which had a size of 64 × 64 pixels, to give it as an input to the model to extract the high-level deep frequency features. We divided the generated data into two folds that included training and testing with a 70:30 % ratio. We utilized 70% of the data for training purposes, and the remaining 30% of the data was used for model validating. We used a 5-folds cross-validation method to test the model generalization. In this method, we used the machine learning library in python to divide our dataset into five-folds and train the models accordingly.

We used a single NVIDIA GeForce GTX 1070 GPU with eight gigabytes onboard memory for the model training and the validating process. We trained the model for 100 epochs with a 0.00001 learning rate and selected decay one unit after 10 epochs. We tried various batch sizes during the model training, but the best result was achieved on 256, which was finally selected for the whole model training process. The Adam optimizer we used in the model training secured the best accuracy for the SER with 0.333 training lost and 0.55 validation lost. Our model architecture is very simple, and it takes less time with training due to its compact model size, which indicates the model’s simplicity and shows the ability for real-time SER applications.” 

  1. How does the proposed model contribute to reducing computational complexity? Why fewer convolutional layers with small respective fields can still learn discriminative features? How to decide the tradeoff between the accuracy and complexity of the proposed method?

Response: Thank you for your interest. We proposed a simple and lightweight convolutional neural network (CNN) architecture with multiple layers using modified kernels and pooling strategy to detect the sensitive cues based on the extraction of the deep frequencies features from the speech spectrograms, which tend to be more discriminative and reliable in speech emotion recognition. In other hand, plain rectangular shape filters for the convolutions and the pooling strategy in order to extract the deep emotional features from the speech spectrogram utilizing frequencies pixels with reliable receptive fields to ensure and enhance the performance of the baseline SER models. For reducing cost computation and the time complexity of the proposed SER model by utilizing fewer convolutional layers with small respective fields in order to learn deep the discriminative features from the speech spectrograms, which was proven in the experimentations. For the convenience of the readers, we documented the statics in experimental section.

  1. L297, the numbers of the references IEMOCAP and EMO-DB are not displayed correctly.

Response: Thank you for your comments and valuable review. In the revised manuscript we proper cite the references. The updated text from the revised version is below for the easy to reviewer.

“We practically proved our system in this section, which we tested over two benchmark IEMOCAP [45] and EMO-DB [46] speech datasets in order to show the significance and the robustness of the model for the SER using speech spectrograms.”

  1. The paper may calculate or list the numbers of the weights or parameters of the proposed CNN model to indicate it’s lightweight.

Response: Thank you for your interest. We added a table that show the parameters of the proposed CNN model, which is indicated the simplicity. The table is added in section 3 after main framework and mention below for easy to reviewer.

Table 1: The overall specification of the proposed model is illustrated with set of convolutions layers, input and output size, and number of units which consist of kernel size, and number of parameters.

Model Layers

Input Tensor

Output Tensor

Parameters

Layer 1, 2, and Layer 3.

3×64×64

10×64×64

L1= 280, L2= 210, and L3= 310

Pooling_1

10×64×64

10×62×64

0

Layer 4, and Layer 5.

10×62×64

20×62×64

L4= 620, and L5= 1220

Pooling_2

20×62×64

20×31×64

0

Layer 6, and Layer 7.

20×31×64

40×31×64

L6= 2440, and L7= 4840

Pooling_3

40×31×64

40×15×64

0

Layer 8.

40×15×64

80×15×64

L8 = 7260

Dense_1

35840

80

2867280

Dense_2

80

30

2430

Softmax

Probabilities

Number of classes

Accordingly

Total parameters = 2974367

Trainable parameters = 2973887

Round 2

Reviewer 1 Report

The authors took care of all my comments and I am satisfied.

Success with future research

Reviewer 2 Report

The authors responded well to my concerns in the last round review. The manuscript has been improved and I think it is acceptable for publication.

Suggestions:

1. In CV, AlexNet is more frequently used than "Alex Net". Please use the former in the paper.

2. Please check the grammar of the paper carefully. Pay attention to the third-person singular form in the whole manuscript.